# Intradermal Fractional ChAdOx1 nCoV-19 Booster Vaccine Induces Memory T Cells: A Follow-Up Study

**DOI:** 10.3390/vaccines12020109

**Published:** 2024-01-23

**Authors:** Ratchanon Sophonmanee, Perawas Preampruchcha, Jomkwan Ongarj, Bunya Seeyankem, Porntip Intapiboon, Smonrapat Surasombatpattana, Supattra Uppanisakorn, Pasuree Sangsupawanich, Sarunyou Chusri, Nawamin Pinpathomrat

**Affiliations:** 1Department of Biomedical Sciences and Biomedical Engineering, Faculty of Medicine, Prince of Songkla University, Songkhla 90110, Thailand; 6510320016@psu.ac.th (R.S.); 6310310128@psu.ac.th (P.P.); 6510320014@psu.ac.th (J.O.); 6210320014@psu.ac.th (B.S.); 2Department of Internal Medicine, Faculty of Medicine, Prince of Songkla University, Songkhla 90110, Thailand; iporntip@medicine.psu.ac.th (P.I.); sarunyouchusri@hotmail.com (S.C.); 3Department of Pathology, Faculty of Medicine, Prince of Songkla University, Songkhla 90110, Thailand; pornapat.s@psu.ac.th; 4Clinical Research Center, Faculty of Medicine, Prince of Songkla University, Songkhla 90110, Thailand; supattra.s@psu.ac.th (S.U.); pasuree.s@psu.ac.th (P.S.)

**Keywords:** intradermal, vaccine, effector T cell, memory T cell, SARS-CoV-2, COVID-19

## Abstract

The administration of viral vector and mRNA vaccine booster effectively induces humoral and cellular immune responses. Effector T cell responses after fractional intradermal (ID) vaccination are comparable to those after intramuscular (IM) boosters. Here, we quantified T cell responses after booster vaccination. ChAdOx1 nCoV-19 vaccination induced higher numbers of S1-specific CD8^+^ memory T cells, consistent with the antibody responses. Effector memory T cell phenotypes elicited by mRNA vaccination showed a similar trend to those elicited by the viral vector vaccine booster. Three months post-vaccination, cytokine responses remained detectable, confirming effector T cell responses induced by both vaccines. The ID fractional dose of ChAdOx1 nCoV-19 elicited higher effector CD8^+^ T cell responses than IM vaccination. This study confirmed that an ID dose-reduction vaccination strategy effectively stimulates effector memory T cell responses. ID injection could be an improved approach for effective vaccination programs.

## 1. Introduction

COVID-19 has gained global recognition for its significant impact on the subsistence of people worldwide. For more than three years, the public has come to understand the seriousness of this infectious disease. As of April 2023, more than 700 million confirmed cases of COVID-19, including approximately 6.9 million deaths, have been reported to the World Health Organization (WHO) (https://covid19.who.int/ accessed on 30 April 2023). To combat this pandemic, the production and global distribution of a variety of COVID-19 vaccines are imperative [1]. Several infectious diseases can be prevented through the effective vaccination. Immunization is intended to prevent specific illnesses and lessen disease effects [2,3]. The unprecedented speed of the development of the COVID-19 vaccine has changed the general strategy for vaccine development. Consequently, safety and efficacy studies were conducted concurrently on the basis of traditional testing protocols [4]. Global efforts to develop and distribute effective vaccines have provided several treatment options [5].

Vaccine effectiveness is also determined by the function of binding antibodies, which prevents SARS-CoV-2 from getting into the cells through humoral immunity mediated by antibodies and memory B cells [6,7]. T cells aid in antibody production and attack infected cells. Cellular immunity, which involves helper CD4^+^ T cells and cytotoxic CD8^+^ T cells, plays a crucial role in the adaptive response to SARS-CoV-2 infection and protects against severe disease [8,9]. Challenges facing COVID-19 vaccination include inequitable vaccine distribution, vaccine hesitancy, waning immunity, and the emergence of variants [10]. The decline in vaccine efficacy results from the emergence of novel variants that are highly transmissible and partially escape neutralizing antibodies [11]. Evidence also indicates that T cell responses are less likely to be affected by spike antigen mutations associated with variants of concern compared to antibody responses [12].

COVID-19 vaccinations cause local reactions such as pain at the injection site, swelling, and soreness of the surrounding area, as well as systemic reactions including headaches, fever, perspiration, shivering, weariness, and exhaustion. Mild symptoms include joint pain, muscular spasms, generalized body aches (myalgia), osteoarticular pain, and back and neck pain. The local side effects include allergic reactions, such as urticarial eruptions and pruritus [13,14]. After the administration of either mRNA or viral vector vaccines, the symptoms become less severe and do not require hospitalization The symptoms persist only a few days after vaccination. Similar finding has been reported in several studies, with most side effects being mild to moderate in severity [15,16].

To minimize these adverse events, intradermal (ID) vaccination is an attractive alternative to conventional COVID-19 vaccine injections. Intradermal injection consistently reduces systemic reactions owing to its mechanism of action [17]. This injection route efficiently stimulates skin-resident dendritic cell-driven T cell-biased conditions by activating B lymphocytes [18]. Reports have demonstrated that one-fifth of the typical dose is safe and well-tolerated when administered intradermally [19]. Therefore, ID administration is an important aspect of a dose-sparing strategy to effectively distribute vaccines. In our previous research on ID vaccination, the immediate adverse reactions most often reported were local reactions; however, fractional ID vaccination significantly reduces systemic reactions compared with full-dose intramuscular (IM) boosters [20,21]. The immune responses obtained from the 1:5 dose of the standard mRNA vaccine ID booster are high but lower than those of the conventional full-dose vaccine booster [20]. Following the booster, antigen-specific IgG levels remain significantly above baseline, and spike-specific T cell responses are also enhanced [21]. Interestingly, a fractional viral vector vaccine booster administered intradermally shows a non-inferior antibody response compared with an additional full dose of IM injection [21]. Effector T cell responses are comparable between both vaccination routes, but the responses are undermined when the interval between the last vaccination and booster administration is prolonged [22,23,24].

Here, we followed-up the participants of previously published trials for 3 months after booster vaccination [20,21]. This study aimed to monitor SARS-CoV-2-specific T cell responses three months after BNT162b2 mRNA or ChAdOx1 nCoV-19 vaccine booster administration following a primary series of inactivated SARS-CoV-2 vaccinations. The memory T cell responses were analyzed based on phenotype and S-1-specific cytokine responses. We showed that fractional ID mRNA or viral vector vaccination enhanced vaccine-induced memory T cell immunity. However, only an ID viral vector booster administered after an extended interval can prolong the longevity of effector memory T cells, which could confer protection against severe disease.

## 2. Materials and Methods

### 2.1. Study Population

The Human Research Ethics Committee granted approval for this study (REC. 64-368-4-1), and this study was conducted under the registered number TCTR20211004001 of the Thai Clinical Trials Registry. All the participants provided written and informed consent. The participants were enrolled as shown in Figure 1. The study set-up can be found in previously published studies [20,21]. Booster vaccination was performed using a plug syringe (TERUMO, KDSS01ST) to minimize the dead space of the vaccine residual. The booster vaccination was performed by experienced research nurses. The immediate and delayed local reactions were observed at 30 min and 7 days after boosting. The vaccine immunogenicity before and after the booster dose was previously reported [18,19]. Antibody responses (anti-receptor-binding domain [RBD] IgG) and vaccine histories were reviewed to exclude participants who were infected or received an extra COVID-19 vaccine during the follow-up period through the “MOHPROMT” database, which was developed by the Thai Ministry of Public Health (Appendix A). Blood samples were collected three months after vaccination for immunological analyses.

### 2.2. Separation of Peripheral Blood Mononuclear Cells (PBMCs)

The blood samples were collected on the day 90 after the 3rd dose. A total of 10 mL of blood was divided split into two heparinized tubes, and the samples were processed within 4–6 h of the blood draw. The blood samples were diluted with RPMI and placed into a SepMATE tube containing Lymphoprep. These samples were then centrifuged at 1200× *g* for 10 min with the brake engaged. The buffy coat layer was poured into a fresh 50 mL tube, supplemented with RPMI, and subjected to a subsequent spin at 300× *g* for 8 min. The resulting cell pellet underwent additional washing with RPMI. Following the final wash, the cell pellet was reconstituted in 3 mL of RPMI for cell counting. The cells were diluted in Trypan blue and enumerated using a counting chamber. The remaining cells were centrifuged at 300× *g* for 8 min and adjusted to a concentration of 3 × 10^6^ PBMCs per milliliter in freezing media (fetal bovine serum (FBS), with 10% DMSO). These cell suspensions were divided into aliquots and placed in Corning CoolCell Containers for an overnight freezing process at −80 °C. Subsequently, the tubes were transferred into liquid nitrogen to await the subsequent investigation.

### 2.3. Flow Cytometry Analysis

The PBMCs were thawed, and R10 (5 mL) was gently added. The cells were rested for 10 min and washed twice in pre-warmed R10 at 37 °C. The cell concentration was adjusted to 2 × 10^6^–3 × 10^6^ PBMCs/mL R10. In a 50 mL tube, 10 μ/mL benzonase was added and incubated for 2 h with loose caps at 37 °C in 5% CO_2_. After incubation, the cells were resuspended in R10. Subsequently, 1 × 10^6^ PBMCs were introduced into a 96-well plate, rinsed with R10, and spun for 5 min at 470× *g* at 22 °C. Each sample underwent stimulation with an S1-peptide pool (ProImmune, Oxford, UK), composed of 15-mers with a 10 amino acid overlap (Appendix A). The peptide pool was thinned to a concentration of 2 μg/mL in R10 supplemented with anti-human CD28 and CD49d. The cells were cultured at 37 °C with 5% CO_2_ for 18 h, with the addition of GolgiPlug (BD Biosciences, NJ, USA) after 2 h. Following stimulation, the plates underwent centrifugation and were washed with PBS. LIVE/DEAD Fixable Aqua Dead Cell Stain (ThermoFisher Scientific, Waltham, MA, USA) was diluted (1:1000) in PBS (Invitrogen, Waltham, MA, USA) and applied to stain the cells for 10 min, followed by a 30 min incubation period with anti-CD3 (BV421), CD4 (APC-H7), CD8 (APC), CD45RO (BV650), and CCR7 (BB515) diluted in 1% bovine serum albumin (BSA) in PBS (FACS buffer) (Appendix A). Following surface staining, the cells underwent fixation and permeabilization using CytoFix (BD Biosciences, NJ, USA) in accordance with the manufacturer’s instructions. The fixed cells were subsequently intracellularly stained with anti-IFN-γ (PC7) and TNF-α (ECD). After staining, the cells were washed with CytoPerm (BD Biosciences, NJ, USA) buffer and reconstituted in FACS buffer for analysis utilizing a CytoFLEX S flow cytometer (Beckman Coulter, Brea, CA, USA). The collected data underwent analysis using FlowJo Software (V10). (FlowJo, Ashland, OR, USA). Gating strategies were used to identify memory T cell populations and cytokine responses (Appendix A). Similarly, broad-stimulation PMA/Ionomycin, a constant-amplitude stimulant, was used to stimulate positive controls to assess the quality and viability of the PBMCs.

### 2.4. Statistical Analysis

Statistical analyses were conducted using GraphPad Prism 9 software (GraphPad Software, Boston, MA, USA). The Kruskal–Wallis’s test, followed by Dunn’s multiple comparisons test, was employed to assess the statistical significance among multiple groups. Statistical significance was acknowledged for values of *p* ≤ 0.05.

## 3. Results

### 3.1. Study Participants

Healthy adults between 18 and 60 years of age (N = 216) who had successfully undergone a two-dose inactivated SARS-CoV-2 vaccination regimen were enrolled (Figure 1). A total of 91 participants who had previously completed primary vaccination over the interval of 8–12 weeks were randomized to receive different regimens of BNT162b2. For the ChAdOx1 nCoV-19 booster, 125 participants within the intervals of 4–8 weeks and >8–12 weeks were enrolled and randomly assigned within each interval to receive different regimens of ChAdOx1 nCoV-19. During this period, Thailand faced restrictions while combating the swift dissemination of the Delta variant of the coronavirus. The Department of Disease Control and the Ministry of Public Health had not officially certified or announced the utilization of mRNA vaccines. Consequently, the strategy for administering booster vaccines after the primary vaccination was restricted to vector vaccines, and the participants with a short duration (4–8 weeks) were eligible only for the vector booster vaccine group. A total of 15 samples from each regimen were used for memory T cell analysis (n = 105). The demographic characteristics of the participants are presented in Table 1. The mean age of the participants was 40 years, and no significant differences were found between the vaccinated groups. For the standard interval, the median time to administering the booster dose (third dose) administration for the conventional interval was 45 days and 73 days for the extended interval.

### 3.2. Effector Cytokine Responses by CD4^+^/CD8^+^ T Cells Induced by BNT162b2 mRNA Vaccine Booster Administration following Primary Series of Inactivated SARS-CoV-2 Vaccinations

To measure the effector cytokine responses (IFN-γ^+^, TNF-α^+^, and IFN-γ^+^TNF-α^+^) of CD4^+^ and CD8^+^ T cells, peripheral blood mononuclear cells (PBMCs) were surface-stained and intracellularly examined using conjugate antibodies (interferon gamma IFN-γ and tumour necrosis factor alpha TNF-α). The stained cells were analyzed using flow cytometry (Figure 2A). The CD4^+^ and CD8^+^ T cell responses exhibited similarity across the study groups (Figure 2B,D). However, no significant differences in the magnitude of effector cytokine responses (IFN-γ^+^, TNF-α+, and IFN-γ^+^TNF-α^+^) were observed in the CD4^+^ T cells three months after BNT162b2 mRNA vaccine booster administration (Figure 2C). A similar trend was observed for the CD8^+^ T cell responses (Figure 2D).

### 3.3. BNT162b2 mRNA Vaccine Booster Administration following Primary Series of Inactivated SARS-CoV-2 Vaccinations Induced S1-Specific CD4^+^/CD8^+^ Effector Memory T Cell (TEM) Responses

To define memory T cell phenotypes three months after BNT162b2 mRNA vaccine booster administration, T cell responses were analyzed using flow cytometry. Surface staining was conducted for the identification of memory CD4^+^ and CD8^+^ T cell populations using CD45RO and CCR7 markers (Figure 3A). Based on gating, memory T cell phenotypes were classified as naïve T cells (T_N_) (CD45RO^−^CCR7^+^), central memory T cells (T_CM_) (CD45RO^+^CCR7^+^), effector memory T cells (T_EM_) (CD45RO^+^CCR7^−^), and effector memory cells re-expressing (T_EMRA_) (CD45RO^−^CCR7^−^) (Figure 3A). The memory phenotypes were examined in both the CD4^+^ and CD8^+^ T cell populations and are shown in pine charts (Figure 3B).

The frequencies of CD4^+^ and CD8^+^ memory T cells (T_CM_) in the group receiving the full IM dose of the BNT162b2 mRNA vaccine were significantly higher than those in the fractional ID dose vaccination group (*p* = 0.0370 and *p* = 0.0024, respectively) (Figure 3C,E). Interestingly, the frequency of CD8^+^ T_EM_ in the half-dose IM group was significantly lower than that in the conventional IM and fractional ID vaccination groups (*p* = 0.0019 and *p* = 0.0356, respectively) (Figure 3E). Following ex vivo stimulation with the S1-peptide pools, no significant change in the CD4^+^ and CD8^+^ memory T cell phenotypes was observed. Next, we measured the S1-specific effector cytokine responses (IFN-γ^+^, TNF-α^+^, and IFN-γ^+^TNF-α^+^) of the CD4^+^ and CD8^+^ T_EM_. Three months after BNT162b2 mRNA vaccine booster administration, the effector cytokine response amplitudes were comparable between the vaccination groups (Figure 3D,F).

### 3.4. ChAdOx1 nCoV-19 Vaccine Booster Administration following Primary Series of Inactivated SARS-CoV-2 Vaccinations Induced S1-Specific CD4^+^/CD8^+^ Effector Memory T Cell (TEM) Responses

In parallel, we observed CD4^+^ and CD8^+^ T cell populations and memory T cell responses to the ChAdOx1 nCoV-19 vaccine booster (Figure 4). The memory T cell phenotypes were identified by gating the expression of CD45RO and CCR7 (Figure 4A). The proportion of memory CD4^+^ T cell phenotypes showed the same trend as the mRNA vaccine booster. Although greater proportions of CD8^+^ T_EM_ were observed with the conventional IM booster as well as with the administration of the fractional ID booster after the extended interval, the differences were not statistically significant (Figure 4B).

To measure IFN-γ^+^, TNF-α^+^, and IFN-γ^+^TNF-α^+^ production by CD4^+^ and CD8^+^ T cells induced by ChAdOx1 nCoV-19 vaccine booster administration, the PBMCs were stained with intracellular conjugate antibodies (IFN-γ and TNF-α) and analyzed using flow cytometry (Figure 5). CD4^+^ and CD8^+^ T cell responses exhibited similarity across the study groups. (Figure 5A,C). IFN-γ^+^-producing CD4^+^ T cell responses to the full IM dose administered after the conventional interval were significantly increased compared with those to the half-dose viral-vector vaccine booster administered after the extended interval (*p* = 0.0196) (Figure 5B). The responses of effector cytokine-producing CD4^+^ T cells in the fractional ID group were significantly higher than those in the half-dose IM booster administered after the same interval of 8–12 weeks (IFN-γ^+^ *p* = 0.0255 and TNF-α^+^ *p* = 0.0097) (Figure 5B). No significant differences were observed in double-positive cytokine secretion (Figure 5B).

The percentage of IFN-γ^+^-producing CD8^+^ T cells produced in response to the full-dose IM viral vector vaccine booster administered after the conventional interval significantly increased compared with the half-dose IM booster administered after the extended interval (*p* = 0.0083) (Figure 5D). The administration of one-fifth of the ID dose showed significantly higher responses than the half-dose IM booster administered after the same interval of 8–12 weeks (*p* < 0.0001) (Figure 5D). The responses to the fractional ID booster administered after a 4–8-week interval remained significantly higher than those to the reciprocal IM booster (*p* = 0.0125) (Figure 5D). Interestingly, the fractional ID booster, administered after an extended interval, elicited the highest responses of TNF-α^+^-producing CD8^+^ T cells compared with other regimes. These included AZ IM half dose administered after 8–12 weeks and AZ ID one-fifth dose, as well as AZ IM full dose administered after a 4–8-week interval (*p* = 0.0068, *p* = 0.0012, and *p* = 0.0029, respectively) (Figure 5D). In addition, this superior ID regimen induced a higher double-positive cytokine-secreting (IFN-γ^+^TNF-α^+^) population compared with the IM booster with half-dose or full-dose administration (*p* ≤ 0.0001 and *p* = 0.0003, respectively) (Figure 5D).

The CD4^+^ and CD8^+^ T_EM_ responses exhibited similarity across the study groups (Figure 5E,G). The S1-specific IFN-γ^+^-producing CD4^+^ T_EM_ responses to the full ID dose administered after the conventional interval were significantly higher compared with those of the half-dose administered IM after an extended interval (*p* = 0.0023) (Figure 5F). By extending the boosting interval, the fractional ID booster significantly enhanced effector cytokine production compared with the cytokine production induced by the half-dose IM booster (IFN-γ^+^; *p* = 0.0053 and TNF-α^+^; *p* = 0.017) (Figure 5F). No significant differences were observed in double-positive cytokine secretion (Figure 5F).

Furthermore, CD8^+^ T_EM_ produced significantly greater antigen-specific IFN-γ^+^ responses after the IM administration of the conventional full-dose booster compared with that after the administration of the fractional IM booster (*p* = 0.0207) (Figure 5H). Again, the fractional ID booster administered after the extended interval significantly enhanced the IFN-γ^+^ CD8^+^ T_EM_ population compared with other reciprocal boosting regimens (AZ, IM, half dose, *p* < 0.0001 and AZ, ID, one-fifth, *p* = 0.0269, respectively (Figure 5H). Consistent with the CD8^+^ T cell responses, the favorable ID booster administered after the extended interval induced the best S1-specific TNF-α^+^ responses by CD8^+^ T_EM_ compared with the other regimes (*p* = 0.0029, *p* = 0.0034, and *p* = 0.0016) (Figure 5H). In addition, significantly higher responses of IFN-γ^+^TNF-α^+^-secreting CD8^+^ T_EM_ were observed in the ID booster group compared with those of the IM booster groups administered half doses or full doses (*p* = 0.0001 and *p* = 0.0003, respectively) (Figure 5H).

Overall, the fractional dose of the ChAdOx1 nCoV-19 vaccine booster administered after an 8–12-week interval provided higher effector cytokine responses compared with those of the other boosting regimens included in this study.

## 4. Discussion

Our findings illustrate the efficacy of the new route of COVID-19 vaccination through ID injection using fractional doses of the BNT162b2 mRNA vaccine or the ChAdOx1 nCoV-19 viral vector vaccine after a primary series of inactivated SARS-CoV-2 vaccinations. ID booster vaccination can increase the longevity of T cell responses compared with the responses observed in conventional vaccination. These findings confirm the appeal of this administration route; ID administration targets the dermal layer, which is rich in immune cells, including dendritic cells, which are crucial for antigen presentation and the initiation of an immune response. ID injection often requires a smaller volume of vaccine, and it can lead to a more targeted and efficient interaction between the vaccine antigen and the immune cells in the skin. This can potentially result in a comparable or even enhanced immune response compared to intramuscular administration. The consideration for intradermal administration in addressing vaccine hesitancy is not just about the injected volume but also about optimizing the interaction between the vaccine and the immune system, potentially providing a more acceptable and effective alternative for those hesitant about traditional intramuscular vaccination.

ID administration increases vaccine uptake among those hesitant to be vaccinated because of the significant concerns associated with the IM administration of COVID-19 vaccines [25,26]. Reviews of influenza vaccines suggest the absence of a substantial difference in the immunogenicity of a fractional ID dose and a full IM dose [4,27,28]. For the COVID-19 vaccination regimens in our previous study, the T cell responses analyzed using flow cytometry on PBMC samples collected from subjects receiving either the BNT162b2 mRNA vaccine or the ChAdOx1 nCoV-19 vaccine booster after 14 and 28 days was not inferior to the intramuscular booster a month after boosting. Therefore, a dose-reduction approach is suggested to increase vaccine coverage. Considering the systemic side effects and limited availability of ID injections of ChAdOx1 nCoV-19 or BNT162b2 mRNA vaccines, the administration of one-fifth of the normal dose is, therefore, a suitable alternative vaccine administration regimen. This route of immunization requires fewer vaccines and effectively stimulates the immune system with minimal systemic side effects [20,21,29].

The effectiveness of the vaccine is significantly influenced by the adaptive immune response, bolstering strong T cell responses that play a crucial role in providing substantial protection against hospitalization [30,31]. Durable immune memory can remain effective for decades, leading to heightened responses and expedited control of pathogens by fostering strong and enduring T cell memories [32,33]. The mRNA used in vaccines encodes for viral spike proteins. These proteins are synthesized in the body after the mRNA, encapsulated in lipid nanoparticles, is injected [34]. The antigens activate immune cells, particularly T helper cells, which release cytokines and stimulate B cells to produce antibodies against the viral spike antigens. Consistent with our findings, we have observed superior anti-RBD IgG responses in the fractional ID AZ vaccination group compared to the conventional IM booster group (Appendix A). This immune response helps in neutralizing the virus. Memory T cells are also generated for future protection. T cytotoxic cells, interacting with MHC-I proteins, produce CD8 proteins that can induce cell death if the cell is infected in the future, enhancing the immune defense against the virus [35,36,37]. The ChAdOx1 nCoV-19 vaccine utilizes a modified chimpanzee DNA adenovirus that has not been previously encountered by humans. This adenovirus serves as a vector to carry genetic instructions for a viral spike protein [38]. Upon injection, the adenovirus binds to host cells, releases its DNA into the cell, and the genetic material migrates to the nucleus. Although it does not integrate into cellular DNA, the host enzymes convert it into mRNA, which moves back into the cytoplasm. The mRNA is then translated by ribosomes into proteins, expressed on cell membranes as MHC-I and MHC-II complexes, prompting cellular and humoral responses against the viral spike protein [35,36]. This could be a significant factor explaining why receiving a viral vaccine as a booster may have a more enduring impact on the immune system compared to receiving an mRNA vaccine.

Our findings show that ID boosters induced antigen-specific effector memory CD4^+^ and CD8^+^ T cells that endured for a minimum of three months post-vaccination. The strength of memory T cell responses hinges on an expanded reservoir of memory T cells that are capable of recognizing pathogen-derived antigens through T cell receptors (TCRs), facilitating a quicker and more robust response upon re-infection [39,40]. TCR recognition occurs via the major histocompatibility complex (MHC) presentation of antigen-presenting cells (APCs), which are more abundant in the skin than in muscles [41]. Resident professional APCs in the skin, called dermal dendritic cells (DDC), can process and present antigens via MHC I and II to CD4^+^ and CD8^+^ T cells, respectively [42,43]. Conventional intramuscular boosters could also induce memory CD4^+^ T cells, but not the CD8^+^ T cell population because fewer APCs are present in the muscle layer than in the skin. CD4^+^ T cells play a central role in promoting effective humoral immunity by providing essential helper signals that influence the differentiation and selection of B cells [44,45]. In addition, CD8^+^ T cells can execute direct antiviral functions by either releasing antiviral cytokines or directly eliminating infected host cells. The crucial contribution of CD4^+^ T cells in fostering the development of potent neutralizing antibody responses to SARS-CoV-2 infection is evident. Due to the viral escape from neutralizing antibodies, a crucial undertaking will be the vaccine stimulation of CD8^+^ T cell responses to provide protection against severe COVID-19 infection [46,47,48].

Our study had a few limitations, as we mainly focused on memory T cell responses. Other T cell phenotypes, such as T cell exhaustion markers, were not characterized after vaccination and boosting because the limited number of collected PBMCs and the inability to fit more markers in the staining made this impossible. We did not evaluate the neutralizing antibodies against the current variants, as this was not within the scope of our study. We expected minimal neutralizing activity from these samples because of the loss of the neutralizing function of the vaccine booster against Omicron variants and later strains. The participants were evaluated for long-term side effects at three months, but no reactions were observed.

Overall, our findings provide insights into the persistence of memory T cell responses to ID BNT162b2 mRNA or ChAdOx1 nCoV-19 viral-vector vaccine booster administration after receiving two doses of the inactivated SARS-CoV-2 vaccine. Populations of helper CD4^+^ and cytotoxic CD8^+^ T_EM_ remained after the booster dose for 90 days. The frequencies of memory T cell phenotypes were comparable between the mRNA and viral vector vaccine boosters. Interestingly, the fractional dose vaccine booster ChAdOx1 nCoV-19 administered after the extended interval induced significantly increased effector cytokine responses (IFN-γ^+^, TNF-α^+^, and IFN-γ^+^TNF-α^+^) by S1-specific helper CD4^+^ and cytotoxic CD8^+^ T_EM_. However, these cell populations were comparable for both routes of vaccination with the BNT162b2 mRNA vaccine. Hence, we demonstrated the longevity of memory T cell responses following the intradermal administration of a fractional dose booster using a viral vector vaccine against COVID-19.

## 5. Conclusions

The production and global distribution of a variety of COVID-19 vaccines are imperative to combat the pandemic. Intradermal vaccination reduces systemic reactions due to its mechanism of action. One-fifth of the typical dose is well-tolerated when administered intradermally; therefore, intradermal administration contributes to a dose-sparing strategy. This study aimed to measure the SARS-CoV-2-specific T cell response three months after BNT162b2 mRNA or ChAdOx1 nCoV-19 vaccine booster administration following a primary series of inactivated SAR-CoV-2 vaccinations. Our findings show that a fractional intradermal dose of ChAdOx1 nCoV-19 elicited higher effector CD8^+^ T cell responses compared with intramuscular vaccination, and that, ultimately, fractional intradermal mRNA or viral vector vaccination enhanced vaccine-induced memory T cell immunity. Intradermal vaccination is a viable alternative in a dose-reduction strategy, resulting in fewer adverse reactions. Our study confirmed that an ID dose-reduction vaccination strategy could be a valuable approach for effective vaccination programs.

## Figures and Tables

**Figure 1 vaccines-12-00109-f001:**
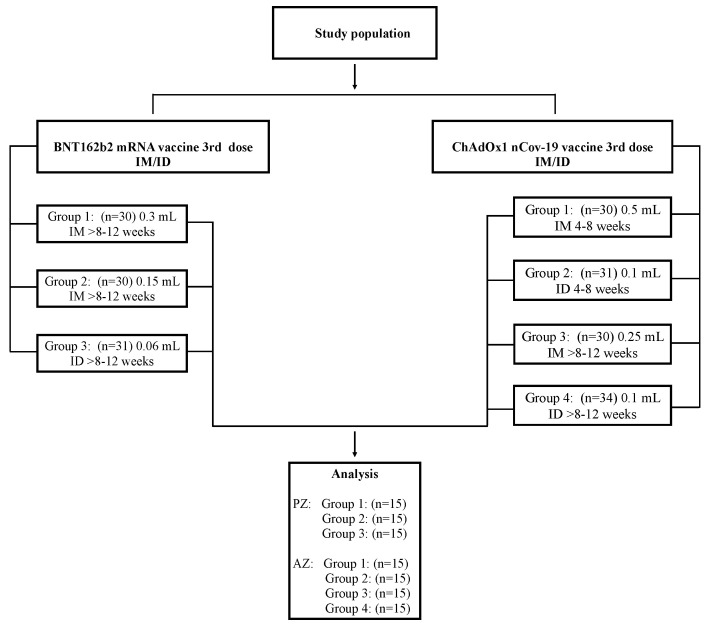
CONSORT chart of the study population. Healthy volunteers ages 18–60 years who had been vaccinated with two doses of inactivated SARS-CoV-2 vaccine for 8–12 weeks were randomized to receive different doses of the BNT162b2 mRNA vaccine (PZ). Thirty participants received a full dose (n = 30, PZ Group 1) or a half dose of the mRNA vaccine intramuscularly (IM) (n = 30, PZ Group 2). The last group received one-fifth of the dose intradermally (ID) (n = 31, PZ Group 3). The ChAdOx1 nCoV-19 vaccine (AZ) was also administered as a booster dose. After completion of a 4–8-week inactivated SARS-CoV-2 vaccination regimen, the participants were randomized to receive a full IM dose of the viral vector vaccine (n = 30, AZ Group 1) or a one-fifth ID dose of AZ (n = 30, AZ Group 2). An interval of >8–12 weeks was also included in the study. Additionally, the participants consented to receive an IM half dose of AZ (n = 30, AZ Group 3) or a one-fifth ID dose of AZ (n = 34, AZ Group 4). Fifteen samples from each group were selected for T cell analysis.

**Figure 2 vaccines-12-00109-f002:**
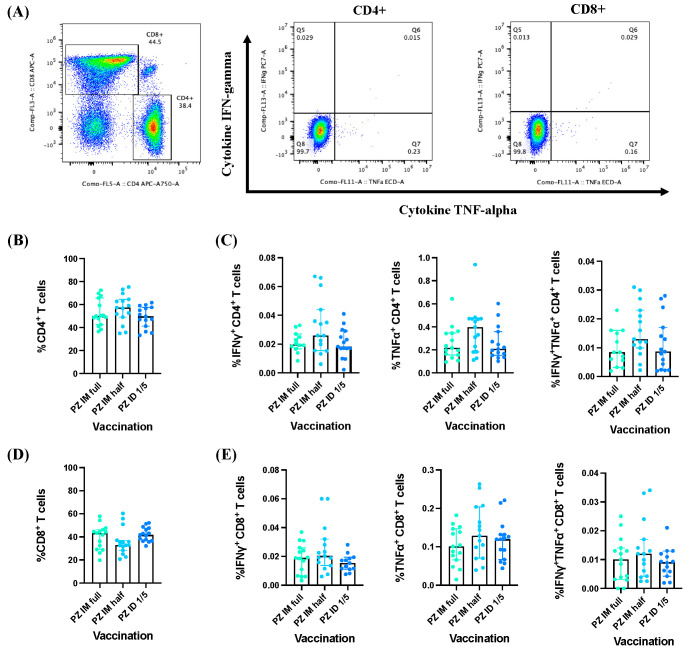
Effector cytokine responses induced by BNT162b2 mRNA vaccine booster administration. (**A**) Gating strategies for selecting CD4^+^ and CD8^+^ T cell populations. (**B**) Percentage of CD4^+^ and (**D**) CD8^+^ T cells. (**C**) Effector cytokine responses (IFN-γ^+^, TNF-α^+^ and IFN-γ^+^TNF-α^+^) of CD4^+^ and (**E**) CD8^+^ T cells. The *p*-value represents the median with a 95% confidence interval (CI). Statistical significance was assessed based on the Kruskal-Wallis test, followed by Dunn’s multiple comparisons test conducted between the vaccinated groups. No significant differences were observed.

**Figure 3 vaccines-12-00109-f003:**
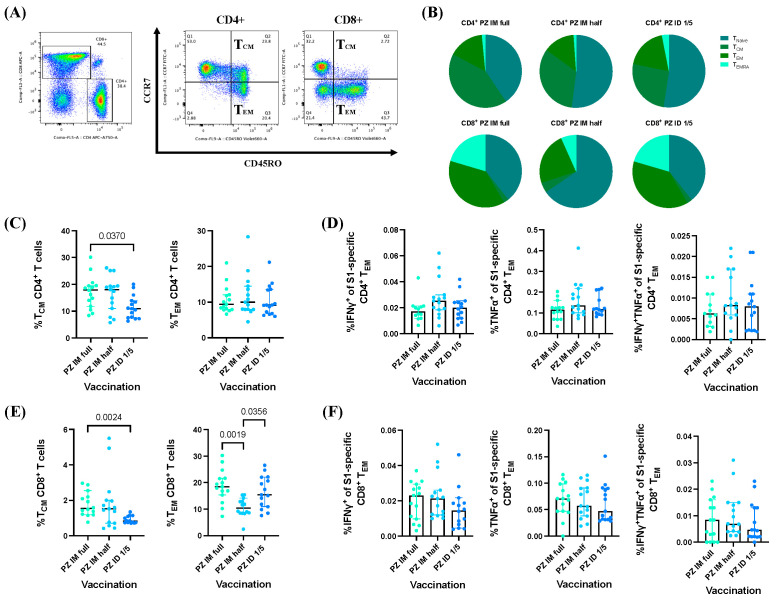
Effector memory T cell (T_EM_) responses after BNT162b2 mRNA vaccine booster administration. (**A**) Gating strategies for selecting CD4^+^ and CD8^+^ T_EM_ populations (CD45RO^+^CCR7^−^). (**B**) The proportion of memory T cell phenotypes presented in pie charts. (**C**) Percentage of S1-specific CD4^+^ and (**E**) CD8^+^ memory T cell phenotypes. (**D**) Effector cytokine responses (IFN-γ^+^, TNF-α^+^, and IFN-γ^+^TNF-α^+^) of S1-specific CD4^+^ and (**F**) CD8^+^ T_EM_ cells. The *p*-value represents the median with a 95% CI. Statistical significance was assessed based on the Kruskal–Wallis test, followed by Dunn’s multiple comparison test between the vaccinated groups.

**Figure 4 vaccines-12-00109-f004:**
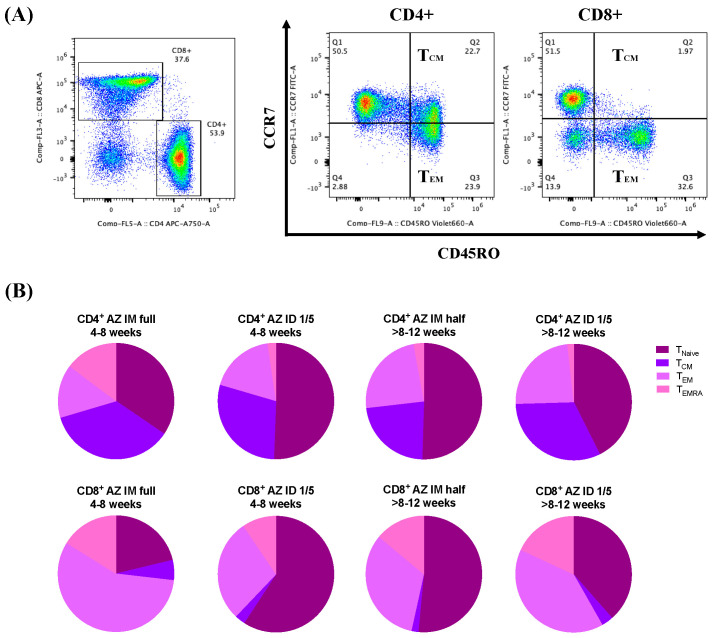
Effector memory T cell (T_EM_) responses after ChAdOx1 nCoV-19 vaccine booster administration. (**A**) Gating strategies for selecting CD4^+^ and CD8^+^ T_EM_ populations (CD45RO^+^CCR7^−^). (**B**) Proportions of memory T cell phenotypes represented in pie charts.

**Figure 5 vaccines-12-00109-f005:**
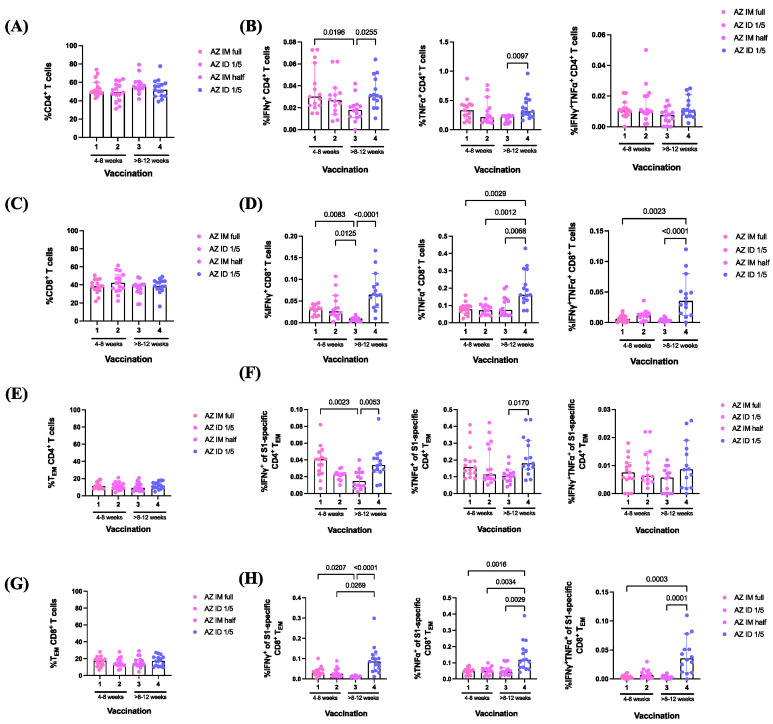
Effector cytokine responses induced by ChAdOx1 nCoV-19 vaccine booster administration. (**A**) Percentage of CD4^+^ and (**C**) CD8^+^ T cells. (**B**) Effector cytokine responses (IFN-γ^+^, TNF-α^+^, and IFN-γ^+^TNF-α^+^) of CD4^+^ and (**D**) CD8^+^ T cells. (**E**) Percentage of CD4^+^ and (**G**) CD8^+^ effector memory T cells (T_EM_). (**F**) Effector cytokine responses (IFN-γ^+^, TNF-α^+^, and IFN-γ^+^TNF-α^+^) of CD4^+^ and (**H**) CD8^+^ T_EM_. The *p*-value represents the median with a 95% CI. Statistical significance was assessed based on the Kruskal–Wallis test, followed by Dunn’s multiple comparisons test conducted between the vaccinated groups.

**Table 1 vaccines-12-00109-t001:** Demographic data of the participants receiving the IM and ID BNT162b2 mRNA vaccine (PZ) and ChAdOx1 nCoV-19 viral vector vaccine (AZ) booster.

Baseline Characteristics	Total n = 105 (%)	G1 PZ IM Full (>8–12 wk.) n = 15 (%)	G2 PZ IM Half (>8–12 wk.) n = 15 (%)	G3 PZ ID 1/5 (>8–12 wk.) n = 15 (%)	G1 AZ IM Full (4–8 wk.) n = 15 (%)	G2 AZ ID 1/5 (4–8 wk.) n = 15 (%)	G3 AZ IM Half (>8–12 wk.) n = 15 (%)	G4 AZ ID 1/5 (>8–12 wk.) n = 15 (%)
Female	68 (64.8%)	11 (73.3)	6 (40)	7 (46.7)	15 (100%)	11 (73.3)	8 (53.3)	10 (66.7)
Mean age, years (IQR)	39.8 (9.05)	41.9 (7.6)	41.7 (8.1)	37.4 (9.2)	33.7 (7.6)	36.1 (8.9)	47.6 (6.4)	40.6 (9.4)
Median duration between primary vaccine series and booster dose, days (IQR)	69.0 (36.0, 94.0)	73 (72, 73)	73 (68.5, 74)	74 (70.5, 74)	45 (44, 46)	51 (45, 52)	68 (67, 73)	80 (76.5, 82)
Median follow-up duration after boosting, days (IQR)	91.0 (88.0, 111)	91 (90.5, 91)	91 (90, 91)	91 (90, 91)	93 (93, 93)	93 (91, 93)	91 (91, 91)	91 (91, 93)

## Data Availability

All the relevant data supporting the findings of this study are available from the corresponding author upon request.

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
