# Peer review of "Intradermal Fractional ChAdOx1 nCoV-19 Booster Vaccine Induces Memory T Cells: A Follow-Up Study"

_vaccines, 2024, doi:10.3390/vaccines12020109_

Round 1

Reviewer 1 Report

Comments and Suggestions for Authors

In this manuscript, Ratchanon Sophonmanee et al quantified T cell responses after booster vaccination. They found that compared to BNT162b2 mRNA vaccination, ChAdOx1 nCoV-19 vaccination induced higher numbers of S1-specific CD8+ memory T cells, consistent with the antibody responses. This study confirmed that an ID dose-reduction vaccination strategy effectively stimulates long-term effector memory T cell responses.

Major comments,

1. The cytokine production are too little when analyse the S1-specific T cells from pan T cells. And S1 stimulation cannot distinguish the whether the cell are S1-specific or bystander effects. Thus, the conclusion is not reliable. The authors need to use S1-loaded tetramer to detect S1-specific T cells and further analyze their function.

2. There is no evidence of long-lived T cells. If the authors want to use this conception, they need to verify these memory T cells are long-lived T cells. 

Author Response

Manuscript ID: vaccines-2794872 - Major Revisions

Reviewer 1

In this manuscript, Ratchanon Sophonmanee et al quantified T cell responses after booster vaccination. They found that compared to BNT162b2 mRNA vaccination, ChAdOx1 nCoV-19 vaccination induced higher numbers of S1-specific CD8+ memory T cells, consistent with the antibody responses. This study confirmed that an ID dose-reduction vaccination strategy effectively stimulates long-term effector memory T cell responses.

Major comments,

  1. The cytokine production are too little when analyse the S1-specific T cells from pan T cells. And S1 stimulation cannot distinguish the whether the cell are S1-specific or bystander effects. Thus, the conclusion is not reliable. The authors need to use S1-loaded tetramer to detect S1-specific T cells and further analyze their function.

We thank reviewer for the thoughtful suggestion. Each sample underwent stimulation with an S1-peptide pool (ProImmune, Oxford, UK). Similarly, broad stimulation PMA/Ionomycin a constant-amplitude stimulant, was used to all the samples to assess the quality and viability of the PBMCs. To observe background of the cytokine responses, unstimulated condition (media) was included in all the samples. Hence, we are assured that the S1 stimulation can distinguish the whether the cells are S1-specific, when compared to the negative control (un-stimulation) and positive control (PMA/Ionomycin stimulation).

  1. There is no evidence of long-lived T cells. If the authors want to use this conception, they need to verify these memory T cells are long-lived T cells. 

We thank reviewer for the thoughtful suggestion. Memory T cells are a subset of T cells that are long-lived. Memory T cells are developed because of the immune system's response to a previous infection or vaccination. They persist in the body after the initial infection has been cleared and provide a rapid and heightened immune response upon re-exposure to the same pathogen. Memory T cells come in two main types: central memory T cells (Tcm) and effector memory T cells (Tem). Central memory T cells primarily reside in lymphoid tissues and can circulate through the bloodstream to secondary lymphoid organs. Effector memory T cells, on the other hand, are found in peripheral tissues and can quickly respond to infections at the site of pathogen entry. The longevity of memory T cells is a crucial aspect of immunological memory. Their ability to persist for extended periods allows the immune system to "remember" and respond more rapidly and effectively to known pathogens during subsequent encounters.

In this study, we have observed T cell populations that expressed memory T cell phenotypes from the PBMCs of vaccinated individuals 90 days after completed the booster dose. These cell populations not only expressed memory T cell markers but can also activated after stimulation with S1 peptide pools which reflect the antigen specific property. We then believe that these cells are long-lived memory T cells at least for 90 days in our setting.

Reviewer 2 Report

Comments and Suggestions for Authors

Review 

This manuscript describes a COVID-19 booster study in which two vaccines (vector and mRNA) were compared in several groups (different doses and different administration routes). The main endpoint was the memory T cell response (phenotypically and functionally). The T-cell response is important for protection against severe disease and possibly also against future variants of concern.

General comments

-        Was COVID-19 vaccination and infection administrated by the ministry of Health, or by the National public health institute, or in a personal booklet? In other words, how can the authors be certain that the booster administered in this study was the third vaccine?

-        Why was immunogenicity after the booster measured after 3 months? Classically the response after a primary series is measured after one month and a booster response is measured after 2 weeks, and then after 3-6-12 months to investigate the (semi-)long term response

-        What were the main in- and exclusion criteria? There is no description of the rationale for the different groups in the main manuscript, also not why different COVID-19 vaccines were given after different time intervals between the primary series and the booster, and why participants were not randomised to receive either the mRNA or the vector vaccine.

-        Blinding is not described

-        Classically ID vaccination is investigated as a dose sparing strategy without losing immunogenicity/ efficacy. The dose sparing aspect is described by the authors describe, but they have not designed the trial as a non-inferiority trial. What was the objective of the study?

-        There is not much experience with intradermal administration of vector vaccines, it would be interesting to briefly report on the safety in the results section of the main manuscript

-        The most striking result of this study is that the fractional dose vaccine booster ChAdOx1nCoV-19 administered after the extended interval induced significantly increased effector cytokine responses (IFN-γ+, TNF-α+, and IFN-γ+TNF-α+) by S1-specific helper CD4+ and cytotoxic CD8+ TEM. The discussion would benefit from a mechanistic hypothesis (why not after a short interval between the primary series and the booster? Why not with IM vaccination and why not with mRNA vaccination?)

-        The authors comment on the fact that they have not measured neutralising antibodies, but the binding antibodies are reported in the supplemental file. Adding these data to the main manuscript would increase the comparability to other COVID-19 vaccine studies

Abstract

-        Line 23: the authors conclude that the T-cell response is long-lived based on measurable cytokines 3 months after vaccination, which is a somewhat optimistic approach of longevity

Introduction

-        Lines 40-41: This sentence (Thus, the safety....) suggests that the safety registration for the COVID-19 vaccines was less vigorously which was not the case, please rephrase

-        Line 54: typo: ...such as painful...

-        Line 59: typo: most symptoms do not require hospitalisation...

-        Lines 65-67: in reference 16 (https://www.ncbi.nlm.nih.gov/pmc/articles/PMC8144901/) there is no mentioning of reduction of IgE production in ID vaccination. Please correct the reference or point out on which data this claim is made

-        Line 84: typo: SARS-CoV-2

Methods

-        How were adverse events monitored, by daily diary, telephone calls, visits? Were symptoms solicited? Please clarify

-        how was infection (COVID-19) monitored: by PCR, anti-N, rapid testing? Medical history? This is always a discussion in COVID-19 vaccine research as it can have a major influence on the results and therefore needs to be described in detail

-        were blood samples collected on more time points within 3 months after the booster in this study? Or was there only one sampling 3 months after the booster? Please describe in more detail

-        The PBMC isolation is described very elaborately. Since this is a standard operation it would benefit from shortening

Results

-        Why were boosters given so shortly after the primary vaccination series?

-        Why were the participants with the short interval (4-8 weeks) only eligible for the vector vaccine booster groups?

-        the authors conclude that no differences are found between the different groups with the PZ booster, but there is a trend towards a higher cytokine response in the PZ IM half group, especially in the CD4 compartment, could this be non-significant due to a small sample size? Or is this a coincidence as one would expect a higher response in the full dose IM (compared to the half dose IM)? Please elaborate

-        the ‘On the other hand’ again suggests a contradiction, but the sentence that follows describes a different comparison (full dose IM vs half dose IM) than the previous sentence (ID vs IM)

Discussion

-        the authors state that vaccine hesitancy could be due to issues with IM administration, which would be resolved with ID administration. Is this because of the injected volume or truly the route of administration? Please elaborate

-        please clarify satisfactory (non-inferior? above a certain cut off?)

-        In intradermal vaccination, dermal dendritic cells are expected to have a bigger role in antigen presentation than Langerhans cells which are in the epidermis

-        Although lower than for previous viral variants, there is still substantial neutralisation of the omicron variant (https://www.nejm.org/doi/full/10.1056/NEJMc2214314

-        Late events were not reported but long-term effects were evaluated. Please specify the difference ‘late’ and ‘long-term’

Figure 1

-        Group 3 PZ: a dose of 0.06mL is very difficult to administer, how do the authors verify that the correct volume was injected?

-        PZ groups: why were these groups only vaccinated IM? (whereas the box above the groups says IM/ID)

-        AZ: why is a different vaccine dose given in groups 1 and 3? There are now 2 variables between these groups that differ (dose and time interval) which hampers drawing a conclusion

Figure 3

-        D and F have ‘S1-specific’ on the Y-axis, does this mean that the other graphs do not depict S1-specific cells?

Figure 5

-        Similar question: are F and H the only S1-specific cells?

Table 1

-        Please clarify which groups were randomized as the demographic variables are significantly different between the different groups (and what does the p-value reflect, a difference between all the groups?)

-        Please clarify ‘median vaccine duration’. Is this the time interval between the last vaccine of the primary series and the booster? Why was this variable compared (considered there is a p-value) if interval between primary series and booster was a criterium to be eligible to different vaccine groups?

- What is ‘median time of vaccine booster’? And why is this significantly different (p<0.001) as this ‘time’ is 91-93 days for all groups which appears to be very comparable

Comments on the Quality of English Language

Minor editing of English language required

Author Response

Manuscript ID: vaccines-2794872 - Major Revisions

Reviewer 2

This manuscript describes a COVID-19 booster study in which two vaccines (vector and mRNA) were compared in several groups (different doses and different administration routes). The main endpoint was the memory T cell response (phenotypically and functionally). The T-cell response is important for protection against severe disease and possibly also against future variants of concern.

General comments,

  1. Was COVID-19 vaccination and infection administrated by the ministry of Health, or by the National public health institute, or in a personal booklet? In other words, how can the authors be certain that the booster administered in this study was the third vaccine?

We thank reviewer for the thoughtful suggestion. The vaccination histories were checked to exclude participants who had been infected or had received additional doses of the COVID-19 vaccine during the follow-up period through the "MOHPROMT" application. This application, developed by the Thai Ministry of Public Health, acts as a central point of communication for accessing COVID-19 services, including vaccination records, laboratory results, and validation of the COVID-19 Digital Health Pass issued by the Thai Government within Thailand. Additionally, we also assessed the antibody levels (anti-receptor-binding domain [RBD] IgG) following the collection of blood samples at day 90. If the antibody responses were higher than the last follow up which could reflect the infection or vaccination during the follow up, the participants would be excluded from the day 90 analysis. Hence, we are assured that the booster vaccine administered in this study is indeed the third dose, and there were no instances of infection throughout the duration of the study.

  1. Why was immunogenicity after the booster measured after 3 months? Classically the response after a primary series is measured after one month and a booster response is measured after 2 weeks, and then after 3-6-12 months to investigate the (semi-)long term response.

We thank reviewer for the suggestion. The In our previous study, we measured the immunogenicity after participant receipt vaccine booster dose at 14, 28 and 90 days. In the setting of the COVID-19 pandemic, it would be very difficult and almost impossible to follow up more than 3 months and be certain about the effect of the booster dose. As per the reviewer’s first question, the participants could infect with the COVID-19 or get additional vaccination during the long-term follow up.

  1. What were the main in- and exclusion criteria? There is no description of the rationale for the different groups in the main manuscript, also not why different COVID-19 vaccines were given after different time intervals between the primary series and the booster, and why participants were not randomised to receive either the mRNA or the vector vaccine.

We thank reviewer for the suggestion. The rationale of different vaccination groups and the time intervals were described as per our previous study. In short, we aimed to observe the immunogenicity after received a fractional dose of a booster using intradermal approach compared to the conventional intramuscular injection with the full dose of the vaccines. The extended interval was included to compare with the convention 4 week-interval. The participants were randomised to receive different vaccination dose/route. The viral vector study was carried out first as it was the main vaccine in Thailand at the time. We then extended our study to look at the immunogenicity of the mRNA vaccine as well when the mRNA vaccine was introduced later in Thailand. Therefore, it was impossible to randomised between the two vaccines.

Due to this study is preliminary study to evaluate the immunogenicity of an intradermal boosted. A sample size of n=30 participants per group has been the minimum sample size necessary to detect a statistically significant difference in immune immunogenicity of an intradermal boosted with 80% power. The full inclusion and exclusion criteria are as follow.

Inclusion criteria

  • Thai adult aged 18-60 years, who have been completed two-dose regimen of inactive SARS-CoV-2 vaccine in the past 1 - 3 months.
  • The subjects are able to and willing to comply with the requirements of the clinical trial program and could complete the 3-month follow-up of the study.
  • Individuals who are in good health condition at the time of entry into the trial as determined by medical history, physical examination and clinical judgment of the investigator and meet the requirements of immunization.
  • The subject can provide with informed consent and sign informed consent form (ICF).

Exclusion criteria

  • Have the medical history or family history of convulsion, epilepsy, encephalopathy, and psychosis.
  • Be allergic to any component of the research vaccines or used to have a history of hypersensitivity or serious reactions to vaccination.
  • Women with positive urine pregnancy test, pregnant or breast-feeding, or have a pregnancy plan within six months.
  • Have acute infectious diseases, including SARS-CoV-2 infection.
  • Have history of SARS-CoV-2 infection
  • Have severe chronic diseases or condition in progress cannot be controlled. For examples, poor controlled DM and uncontrolled HT.
  • Have the history of urticaria 1 year before receiving the investigational vaccine.
  • Have known underlying diseases of thrombocytopenia or other coagulation disorders (which may cause contraindications for intramuscular injection).
  • Have needle sickness.
  • Have the history of immunosuppressive therapy, cytotoxic therapy, or systemic corticosteroids.
  • Have received blood products within 4 months before injection of investigational vaccines.
  • Under anti-tuberculosis treatment.
  • Not be able to follow the protocol, or not be able to understand the informed consent according to the researcher's judgment, due to various medical, psychological, social, or other conditions.
  1. Blinding is not described.

We thank reviewer for the observation. We indeed did the blinding. The participants did not know the dose of the vaccine that was injected. Of course, the route of vaccination cannot be blinded. The T cell analysis was not blinded. However, we randomly selected 15 samples per vaccination group to be tested for memory T cell responses.

  1. Classically ID vaccination is investigated as a dose sparing strategy without losing immunogenicity/ efficacy. The dose sparing aspect is described by the authors describe, but they have not designed the trial as a non-inferiority trial. What was the objective of the study?

We thank reviewer for the observation. The objective of the study is intradermal BNT162b2 mRNA or ChAdOx1 nCoV-19 boosters after 2 inactivated SARS-CoV-2 vaccinations enhance better memory T-cell responses compared to the conventional intramuscular booster at 3 months. This study is only focused on memory T cell analysis. The clinical trial design was described in the previous publication.

Pinpathomrat, Nawamin, et al. "Immunogenicity and safety of an intradermal ChAdOx1 nCoV-19 boost in a healthy population." npj Vaccines 7.1 (2022): 52.

Intapiboon, Porntip, et al. "Immunogenicity and safety of an intradermal BNT162b2 mRNA vaccine booster after two doses of inactivated SARS-CoV-2 vaccine in healthy population." Vaccines 9.12 (2021): 1375.

  1. There is not much experience with intradermal administration of vector vaccines, it would be interesting to briefly report on the safety in the results section of the main manuscript.

We thank reviewer for the suggestion. The safety data of intradermal administration of the vector vaccine has been reported in our previous publication.

Pinpathomrat, Nawamin, et al. "Immunogenicity and safety of an intradermal ChAdOx1 nCoV-19 boost in a healthy population." npj Vaccines 7.1 (2022): 52.

  1. The most striking result of this study is that the fractional dose vaccine booster ChAdOx1nCoV-19 administered after the extended interval induced significantly increased effector cytokine responses (IFN-γ+, TNF-α+, and IFN-γ+TNF-α+) by S1-specific helper CD4+ and cytotoxic CD8+ TEM. The discussion would benefit from a mechanistic hypothesis (why not after a short interval between the primary series and the booster? Why not with IM vaccination and why not with mRNA vaccination?)

We thank reviewer for the helpful suggestion. Our mechanistic hypothesis on this observation is T cell exhaustion. Experiencing T cell exhaustion can result from receiving an excessively reactive booster vaccine following the initial vaccine dose. Short interval between vaccination doses, high dose of the vaccine and conventional IM injection can enhance very strong immune responses immediately after the injection. T-cell exhaustion refers to a state of dysfunction that can occur in T cells during prolonged exposure to antigen stimulation, particularly in the context of chronic infections or cancer. Exhausted T cells exhibit reduced effector functions and diminish proliferative capacity, leading to an impaired ability to combat the target pathogen or cancer cells. This phenomenon is often characterized by the upregulation of inhibitory receptors, such as PD-1 (programmed cell death protein 1). The presence of these inhibitory receptors contributes to the T cells' inability to mount an effective immune response.

  1. The authors comment on the fact that they have not measured neutralising antibodies, but the binding antibodies are reported in the supplemental file. Adding these data to the main manuscript would increase the comparability to other COVID-19 vaccine studies.

We thank reviewer for the suggestion. Explaining the results of antibody levels in the main manuscript is challenging, given our primary emphasis on investigating T-cell immune responses. However, we have mentioned and discussed the antibody responses in the discussion.

Abstract,

  1. Line 23: the authors conclude that the T-cell response is long-lived based on measurable cytokines 3 months after vaccination, which is a somewhat optimistic approach of longevity.

We thank reviewer for the suggestion. We have revised as per reviewer’s comment.

Introduction,

  1. Lines 40-41: This sentence (Thus, the safety....) suggests that the safety registration for the COVID-19 vaccines was less vigorously which was not the case, please rephrase.

We thank reviewer for the suggestion. We have revised as per reviewer’s comment.

  1. Line 54: typo: ...such as painful...

We thank reviewer for the suggestion. We have revised as per reviewer’s comment.

  1. Line 59: typo: most symptoms do not require hospitalisation...

We thank reviewer for the suggestion. We have revised as per reviewer’s comment.

  1. Lines 65-67: in reference 16 (https://www.ncbi.nlm.nih.gov/pmc/articles/PMC8144901/) there is no mentioning of reduction of IgE production in ID vaccination. Please correct the reference or point out on which data this claim is made.

We thank reviewer for the suggestion. We have revised as per reviewer’s comment.

  1. Line 84: typo: SARS-CoV-2

We thank reviewer for the suggestion. We have revised as per reviewer’s comment.

Methods,

  1. How were adverse events monitored, by daily diary, telephone calls, visits? Were symptoms solicited? Please clarify.

We thank reviewer for the question. Solicited local adverse reactions at 30 min and 7 days after boosting. The immediate and delayed local reactions were observed after injection. Seven days after boosting, local adverse events were recorded for comparing between booster groups. Our adverse event monitoring report has been published. (Intapiboon, et al., 2021 and Pinpathomrat, et al., 2022)

  1. how was infection (COVID-19) monitored: by PCR, anti-N, rapid testing? Medical history? This is always a discussion in COVID-19 vaccine research as it can have a major influence on the results and therefore needs to be described in detail.

We thank reviewer for the question. As previously elucidated earlier. The vaccination histories were checked to exclude participants who had been infected or had received additional doses of the COVID-19 vaccine during the follow-up period through the "MOHPROMT" database which recorded by the Thai Ministry of Public Health.

  1. were blood samples collected on more time points within 3 months after the booster in this study? Or was there only one sampling 3 months after the booster? Please describe in more detail.

We thank reviewer for the question. In this study, blood samples were collected on the day of vaccination, and then 14, 28 and 90 days after the booster for immunological analysis. The results of the study have been published. (Intapiboon, et al., 2021, Pinpathomrat, et al., 2022 and Sophonmanee, et al., 2022)

  1. The PBMC isolation is described very elaborately. Since this is a standard operation it would benefit from shortening

We thank reviewer for the suggestion. We have revised as per reviewer’s comment.

Results,

  1. Why were boosters given so shortly after the primary vaccination series?

We thank reviewer for the question. The Department of Disease Control and the Thai Ministry of Public Health has released the guidelines for people who have never received the COVID-19 vaccine should complete the vaccination immediately. Those who have already completed the primary vaccination series should get a booster dose by keeping the interval around 4 weeks or later than that after received primary vaccination. This occurred because, during that period, Thailand was grappling with the swift dissemination of the Delta variant of the coronavirus and people were living with fear of getting infected.     

  1. Why were the participants with the short interval (4-8 weeks) only eligible for the vector vaccine booster groups?

We thank reviewer for the question. During that period, the Thai Ministry of Public Health had not yet obtained certification and officially declared the utilization of the mRNA vaccine. The strategy for dispensing booster vaccines after the primary vaccination series was consequently limited to viral-vector vaccines.

  1. the authors conclude that no differences are found between the different groups with the PZ booster, but there is a trend towards a higher cytokine response in the PZ IM half group, especially in the CD4 compartment, could this be non-significant due to a small sample size? Or is this a coincidence as one would expect a higher response in the full dose IM (compared to the half dose IM)? Please elaborate.

We thank reviewer for the question. A higher cytokine response was observed in the PZ IM half dose group, especially in the CD4 compartment, although not statistically significant. This result again suggests that fractional dose could be beneficial in keeping memory T cell responses on long-term compared to the conventional full dose vaccination. We believe this could explain the hypothesis of T cell exhaustion after a potent booster shot.

  1. the ‘On the other hand’ again suggests a contradiction, but the sentence that follows describes a different comparison (full dose IM vs half dose IM) than the previous sentence (ID vs IM)

We thank reviewer for the suggestion. We have revised as per reviewer’s comment.

Discussion,

  1. the authors state that vaccine hesitancy could be due to issues with IM administration, which would be resolved with ID administration. Is this because of the injected volume or truly the route of administration? Please elaborate.

We thank reviewer for the question. The IM vaccination route consistently show the reduction of systemic side effect compared to IM route. IM administration involves injecting the vaccine into the muscle tissue, whereas ID administration targets the dermal layer of the skin. The skin, particularly the dermis, is rich in immune cells, including dendritic cells, which are crucial for antigen presentation and the initiation of an immune response. The choice between IM and ID administration is not solely about the volume injected but also about the type of immune response elicited. ID administration often requires a smaller volume of vaccine, and it can lead to a more targeted and efficient interaction between the vaccine antigen and the immune cells in the skin. This can potentially result in a comparable or even enhanced immune response compared to IM administration. Therefore, the consideration for ID administration in addressing vaccine hesitancy is not just about the injected volume but also about optimizing the interaction between the vaccine and the immune system, potentially providing a more acceptable and effective alternative for those hesitant about traditional IM vaccination.

  1. please clarify satisfactory (non-inferior? above a certain cut off?)

We thank reviewer for the suggestion. We have revised as per reviewer’s comment.

  1. In intradermal vaccination, dermal dendritic cells are expected to have a bigger role in antigen presentation than Langerhans cells which are in the epidermis.

We thank reviewer for the suggestion. We have revised as per reviewer’s comment.

  1. Although lower than for previous viral variants, there is still substantial neutralisation of the omicron variant (https://www.nejm.org/doi/full/10.1056/NEJMc2214314

We thank reviewer for the suggestion. We have revised as per reviewer’s comment.

  1. Late events were not reported but long-term effects were evaluated. Please specify the difference ‘late’ and ‘long-term’

We thank reviewer for the suggestion. Late vaccine adverse events were not reported as there were no events noticed by the participants. Our study then analysed and reported long-term T cell responses at day 90.

Figure 1,

  1. Group 3 PZ: a dose of 0.06mL is very difficult to administer, how do the authors verify that the correct volume was injected?

We thank reviewer for the suggestion. We took this matter into consideration. We used a special syringe that had no dead-space. Only experienced research nurses performed the injection.

  1. PZ groups: why were these groups only vaccinated IM? (whereas the box above the groups says IM/ID)

We thank reviewer for the suggestion. We have revised as per reviewer’s comment.

  1. AZ: why is a different vaccine dose given in groups 1 and 3? There are now 2 variables between these groups that differ (dose and time interval) which hampers drawing a conclusion.

We thank reviewer for the suggestion. Because the IM half dose of AZ (n = 30, AZ Group 3) constitutes an additional collection sample after the collection of samples in other study groups was completed.

Figure 3,

  1. D and F have ‘S1-specific’ on the Y-axis, does this mean that the other graphs do not depict S1-specific cells?

We thank reviewer for the suggestion. In Fig3D and F, we measured S1 specific cytokines after re-stimulated with S1 peptide pools. Other graphs in Fig3 showed phenotypes of the T cells.

Figure 5,

  1. Similar question: are F and H the only S1-specific cells?

We thank reviewer for the suggestion. In Fig5F and h, we measured S1 specific cytokines after re-stimulated with S1 peptide pools. Other graphs in Fig5 showed phenotypes of the T cells.

Table 1,

  1. Please clarify which groups were randomized as the demographic variables are significantly different between the different groups (and what does the p-value reflect, a difference between all the groups?)

We thank reviewer for the question. Volunteers aged 18–60 years who had been vaccinated with two doses of inactivated SARS-CoV-2 vaccine for 8–12 weeks were randomised to receive different doses of the BNT162b2 mRNA vaccine (PZ). Thirty participants received a full dose (n = 30, PZ Group 1) or a half dose of the mRNA vaccine intramuscularly (IM) (n = 30, PZ Group 2). The last group received one-fifth of the dose intradermally (ID) (n = 31, PZ Group 3). The ChAdOx1 nCoV-19 vaccine (AZ) was also administered as a booster dose. After completion of a 4–8-week inactivated SARS-CoV-2 vaccination regimen, participants were randomised to receive a full IM dose of the viral vector vaccine (n = 30, AZ Group 1) or a one-fifth ID dose of AZ (n = 30, AZ Group 2). An interval of >8–12 weeks was also included in the study. Additionally, participants consented to receive an IM half dose of AZ (n = 30, AZ Group 3) or a one-fifth ID dose of AZ (n = 34, AZ Group 4) which is overseen by skilled nursing professionals.

  1. Please clarify ‘median vaccine duration’. Is this the time interval between the last vaccine of the primary series and the booster? Why was this variable compared (considered there is a p-value) if interval between primary series and booster was a criterium to be eligible to different vaccine groups?

We thank reviewer for the question. Median vaccine duration is a median of vaccine duration of the time interval between the last vaccine of the primary series and the booster.

  1. What is ‘median time of vaccine booster’? And why is this significantly different (p<0.001) as this ‘time’ is 91-93 days for all groups which appears to be very comparable.

We thank reviewer for the question. Median time of vaccine booster is the period from when the volunteer received the booster vaccine until the date of sample collection at 3 months or approximately 90 days. The statistical analysis that we used was the Kruskal Wallis’s test which compared more that 2 groups. The groups had the median of 93 would be significantly different compared to the groups that had the median of 91. In each group had around 30 participants which repetitively had the duration of 90 and 91 days. By having 93 days of the interval would make it significantly different. 

Round 2

Reviewer 1 Report

Comments and Suggestions for Authors

1. I have not seen any positive control in the manuscript. After production, cytokines are quickly secreted out of the cell. The authors did not mention the use of Golgi inhibitors, therefore the proportion of positive intracellular cytokine staining is very low, and these results may not necessarily reflect the actual situation.

2.  I still oppose the use of the term 'long-lived memory T cells' as the authors have not provided any evidence to confirm the lifespan of these memory T cells.

Author Response

Reviewer 1

We thank reviewer 1 for your valuable comments.

  1. I have not seen any positive control in the manuscript. After production, cytokines are quickly secreted out of the cell. The authors did not mention the use of Golgi inhibitors, therefore the proportion of positive intracellular cytokine staining is very low, and these results may not necessarily reflect the actual situation.

Intramuscular full dose vaccination was included as a regimen control. In the intracellular cytokine analysis, PMA/Ionomycin stimulation was included as a positive control to check the cell quality and cytokine responses after being broadly stimulated. Glolgi plug was added after 2 hr of ex-vivo stimulation to stop the cytokine to secret out of the cells. We have revised the methods to be clear on the cytokine analysis in section 2.3. Flow cytometry analysis.

  1. I still oppose the use of the term 'long-lived memory T cells' as the authors have not provided any evidence to confirm the lifespan of these memory T cells.

We have revised not to use the word 'long-lived’. We’ve corrected to state the cell population as ‘memory T cells' which were confirmed by their memory phenotypes.

Reviewer 2 Report

Comments and Suggestions for Authors

After the second review it became clear that this is a follow-up study of a previously publishes study. Therefore, the manuscript would benefit greatly from shortening, and I would suggest to make a short communication/ brief report with the most important results. Also, the authors should clearly write that this is a follow-up study.

Several previous points have been adjusted accordingly, but several others have not. Please find below the specific comments to the author's answers.

2.       If I understand correctly, this is the second publication of the same cohort, but now with a longer follow-up period. Is that correct? Then the authors should be more clear on this point and refer to the previous publication in the title (e.g. follow-up study for 3 months cellular immune responses to vaccination ….) and in the introduction. And then it makes sense not to publish the 14 day and 28 days post-booster responses again.

3.       If indeed this cohort has been published previously the authors can refer to that publication for the in- and exclusion criteria, and the set-up of the study

4.       Why was laboratory personnel not blinded for the treatment group? That is standard procedure

5.       The authors refer to a publication in NPJ Vaccines, but is the current study not a follow-up study of https://www.mdpi.com/2076-393X/10/9/1494?

8.       Why is it challenging to discuss antibody results? Then I propose not to mention them at all

9.       This is not adjusted in the abstract

10.   Still a bit worrisome, compression of traditional testing protocol sounds as though safety protocols were not taken very seriously

12.   No change has been made

13.   The authors changed IgE to IgG, but in reference 16 there is no mentioning of IgG in reference IgG

16.   This is not an answer to the question. The authors use an antibody level to account for breakthrough infections, but this is not valid as antibody levels can vary widely

Figure 1.

28. which syringe has no dead space? Please mention this in materials and methods. Many researchers would be interested in this

Figure 3 and 5

31 and 32 then why is there still the word ‘cells’ on the y-axis when it should be cytokines?

Table 1.

34. this is not altered in the table (still not clear)

35. text has not been altered to make it more clear, and it remains very unlikely that there is a difference between these groups with similar time intervals

Author Response

Reviewer 2

 We thank reviewer 2 for your time and suggestions reviewing this manuscript.

After the second review it became clear that this is a follow-up study of a previously publishes study. Therefore, the manuscript would benefit greatly from shortening, and I would suggest to make a short communication/ brief report with the most important results. Also, the authors should clearly write that this is a follow-up study.

Several previous points have been adjusted accordingly, but several others have not. Please find below the specific comments to the author's answers.

  1. If I understand correctly, this is the second publication of the same cohort, but now with a longer follow-up period. Is that correct? Then the authors should be more clear on this point and refer to the previous publication in the title (e.g. follow-up study for 3 months cellular immune responses to vaccination ….) and in the introduction. And then it makes sense not to publish the 14 day and 28 days post-booster responses again.

      We are sorry for did not be clearer on the standing point of this manuscript. We have revised the title which is now “Intradermal fractional ChAdOx1 nCoV-19 booster vaccine induces memory T cells, a follow-up study”. The introduction has been revised to mention that it is a follow up study of previously published work. In the last two paragraphs,

  • In our previous research on ID vaccination, the immediate adverse reactions most often reported were local reactions; however, fractional ID vaccination significantly reduces systemic reactions compared with full-dose intramuscular (IM) boosters (18, 19).
  • we followed-up the participants for 3 months after booster vaccination of previously published trials (18, 19).
  1. If indeed this cohort has been published previously the authors can refer to that publication for the in- and exclusion criteria, and the set-up of the study

We have referred to our previously published articles in the methods as suggested.

  • The study set-up can be found the previously published study (18, 19). The immediate and delayed local reactions were observed at 30 minutes and 7 days after boosting. The vaccine immunogenicity before and after the booster dose was previously reported (18, 19).
  1. Why was laboratory personnel not blinded for the treatment group? That is standard procedure

We did blind the laboratory personnel in our group until the immunogenicity analysis was done for day 14 and day 28 after booster. This study is a follow up study. Therefore, the same set of the laboratory personnel worked on the sample unblind for day 90 study.

  1. The authors refer to a publication in NPJ Vaccines, but is the current study not a follow-up study of https://www.mdpi.com/2076-393X/10/9/1494?

This study is a follow up study of the 2 articles followed.

  1. Intapiboon, P. et al. Immunogenicity and safety of an intradermal BNT162b2 mRNA vaccine booster after two doses of inactivated SARS-CoV-2 vaccine in healthy population. Vaccines 9, 1375 (2021).
  2. Pinpathomrat, N. et al. Immunogenicity and safety of an intradermal ChAdOx1 nCoV-19 boost in a healthy population. NPJ Vaccines 7, 52 (2022).

https://www.mdpi.com/2076-393X/10/9/1494 reported T cell analysis of the first paper (https://www.mdpi.com/2076-393X/9/12/1375)

  1. Why is it challenging to discuss antibody results? Then I propose not to mention them at all

      Antibody analysis is not our aim. We evaluated anti-RBD IgG just to exclude previous infection and vaccination during the follow-up. This work is focused on Memory T cell study 90 days after the booster vaccination.

  1. This is not adjusted in the abstract

We have now revised to move the word “long-lived” and “longevity” from the abstract as well as the title. 

  1. Still a bit worrisome, compression of traditional testing protocol sounds as though safety protocols were not taken very seriously

      We have revised as “Consequently, safety and efficacy studies were conducted concurrently on the basis of traditional testing protocols”.

  1. No change has been made

      We have revised as “After the administration of either mRNA or viral vector vaccines, the symptom become less severe and do not require hospitalisation.”

  1. The authors changed IgE to IgG, but in reference 16 there is no mentioning of IgG in reference IgG

We have revised as “this injection route efficiently stimulates skin-resident dendritic cell-driven T cell biased conditions by activating B lymphocytes”

The main text in the cited paper is “DDCs capture antigens deposited in the dermis and migrate to regional lymph nodes, where the antigens are presented to T lymphocytes. Soluble antigens also migrate to lymph nodes, resulting in activation of B lymphocytes. Due to the abundance of APCs in the dermis, ID administration of reduced antigenic doses (most often 20% or 30% of the standard amount of antigen), can induce immune responses equivalent to standard doses administered IM or SC.”

  1. This is not an answer to the question. The authors use an antibody level to account for breakthrough infections, but this is not valid as antibody levels can vary widely

We understand that the antibody responses can vary widely. However, we have recorded the antibody level at 1 month compared to 3 months after the booster. During these two months of follow up, if the participant exposure to COVID-19 infection or vaccination, the antibody level will higher than the baseline recorded at 1 month. We cannot think of the case that would make the antibody level sustain or less than the baseline after being infected or vaccinated in less than 2 months.

We do agree that testing for COVID-19 infection using antigen test would be a clear evident to prove the recent or current infection but that would not provide a good evident of previous infection earlier than that. Testing for anti-N would provide information of previous infection. However, anti-N is not positive in every case and the duration of its persistent is very variable.

Figure 1. 

  1. which syringe has no dead space? Please mention this in materials and methods. Many researchers would be interested in this

We have added the syringe information in the method section which is TERUMO Syringe KDSS01ST 1mL with 25Gx1”needle. Attached the photos of the syringe and needle we used below. The syringe comes with a plug filling the dead space inside the syringe-needle connection.

Figure 3 and 5

31 and 32 then why is there still the word ‘cells’ on the y-axis when it should be cytokines?

Fig3C E and Fig5A C D G showed percentage of Tem and Tcm cells, not the cytokines. Fig3D F and Fig5 B D F H showed cytokine producing cells gated from Tem parent, not just the percentage of cytokine.

Table 1.

  1. this is not altered in the table (still not clear)

We have amended in the table as “Median of vaccine duration between primary series and the booster dose, day (IQR)

  1. text has not been altered to make it more clear, and it remains very unlikely that there is a difference between these groups with similar time intervals

We have revised in the table as “Median of follow-up duration after boosting, day (IQR)We also have removed the p-value from multiple comparison to avoid the confusion that may cause.
